# Increased Extracellular Adenosine in Radiotherapy-Resistant Breast Cancer Cells Enhances Tumor Progression through A2AR-Akt-β-Catenin Signaling

**DOI:** 10.3390/cancers13092105

**Published:** 2021-04-27

**Authors:** Hana Jin, Jong-Sil Lee, Dong-Chul Kim, Young-Shin Ko, Gyeong-Won Lee, Hye-Jung Kim

**Affiliations:** 1Department of Pharmacology, College of Medicine, Institute of Health Sciences, Gyeongsang National University, Jinju 52727, Korea; hanajin.kr@daum.net (H.J.); shini33@naver.com (Y.-S.K.); 2Department of Pathology, College of Medicine, Institute of Health Sciences, Gyeongsang National University Hospital, Gyeongsang National University, Jinju 52727, Korea; jongsil25@gnu.ac.kr (J.-S.L.); kdcjes@hanmail.net (D.-C.K.); 3Division of Hematology-Oncology, Department of Internal Medicine, College of Medicine, Institute of Health Sciences, Gyeongsang National University Hospital, Gyeongsang National University Jinju, Jinju 52727, Korea

**Keywords:** adenosine, A2AR, Akt, β-catenin, radiotherapy-resistant breast cancer cells

## Abstract

**Simple Summary:**

In our previous study, purinergic P2Y_2_ receptor (P2Y_2_R) activation by ATP was found to play an important role in tumor progression and metastasis by regulating various responses in cancer cells and modulating crosstalk between cancer cells and endothelial cells (ECs). Therefore, we expected that P2Y_2_R would play a critical role in radioresistance and enhanced tumor progression in radioresistant triple-negative breast cancer (RT-R-TNBC). However, interestingly, P2Y_2_R expression was slightly decreased in RT-R-TNBC cells, while the expression of A2AR was significantly increased both in RT-R-TNBC cells and in tumor tissues, especially triple negative breast cancer (TNBC) tissues of breast cancer (BC) patients. Thus, we aimed to investigate the role of adenosine A2A receptor (A2AR) and its signaling pathway in the progression of RT-R-TNBC. The results reveal for the first time the role of A2AR in the progression and metastasis of RT-R-BC cells and suggest that the adenosine (ADO)-activated intracellular A2AR signaling pathway is linked to the AKT-β-catenin pathway to regulate RT-R-BC cell invasiveness and metastasis.

**Abstract:**

Recently, we found that the expressions of adenosine (ADO) receptors A2AR and A2BR and the ectonucleotidase CD73 which is needed for the conversion of adenosine triphosphate (ATP) to adenosine diphosphate (ADP) and the extracellular ADO level are increased in TNBC MDA-MB-231 cells and RT-R-MDA-MB-231 cells compared to normal cells or non-TNBC cells. The expression of A2AR, but not A2BR, is significantly upregulated in breast cancer tissues, especially TNBC tissues, compared to normal epithelial tissues. Therefore, we further investigated the role of ADO-activated A2AR and its signaling pathway in the progression of RT-R-TNBC. ADO treatment induced MDA-MB-231 cell proliferation, colony formation, and invasion, which were enhanced in RT-R-MDA-MB-231 cells in an A2AR-dependent manner. A2AR activation by ADO induced AKT phosphorylation and then β-catenin, Snail, and vimentin expression, and these effects were abolished by A2AR-siRNA transfection. In an in vivo animal study, compared to 4T1-injected mice, RT-R-4T1-injected mice exhibited significantly increased tumor growth and lung metastasis, which were decreased by A2AR-knockdown. The upregulation of phospho-AKT, β-catenin, Snail, and vimentin expression in mice injected with RT-R-4T1 cells was also attenuated in mice injected with RT-R-4T1-A2AR-shRNA cells. These results suggest that A2AR is significantly upregulated in BC tissues, especially TNBC tissues, and ADO-mediated A2AR activation is involved in RT-R-TNBC invasion and metastasis through the AKT-β-catenin pathway.

## 1. Introduction

Various components of the tumor microenvironment (TME), including adipocytes, fibroblasts, macrophages, and neutrophils [1], interact with breast cancer (BC) cells directly and/or indirectly, promoting stemness, resistance to chemotherapy and/or radiotherapy, and metastatic and invasion potential. ATP is one of the main components of the TME, and it regulates tumor growth and progression and the antitumor immune response [2,3]. Our previous study also showed that MDA-MB-231 cells, triple-negative breast cancer (TNBC) cells that show highly metastatic properties, release higher levels of ATP and show greater P2Y purinoceptor 2 (P2Y_2_R) activity than non-TNBC cells with low metastatic potential, such as MCF-7 cells; the study also showed that P2Y_2_R activation by ATP plays an important role in tumor progression and metastasis by regulating various responses in cancer cells and modulating crosstalk between cancer cells and endothelial cells (ECs) [4].

Extracellular adenosine (ADO) is a purine nucleoside and is mainly generated by the breakdown of ATP by two ectonucleotidases, CD39 and CD73 [5,6]. Recently, ADO has been a topic of interest in cancer studies and is one of the multiple constituents in the TME that affects host and tumor responses [7]. Abundant extracellular ADO can strongly propagate various signaling pathways through P1 receptor activation, which regulates the properties and functions of various cells, including tumor cells and immune cells, in the TME [3]. ADO mediates intracellular responses by activating the G-protein-coupled P1 receptor and A1R, A2AR, A2BR, and A3R, and each receptor exhibits distinct cell and tissue distributions, pharmacological properties, and secondary effector signaling patterns [8]. Both A1R and A3R couple to inhibitory G-proteins G_i_ or G_o_ resulting in a reduction in cAMP accumulation via adenyl cyclase (AC) inhibition [9], while A2AR and A2BR couple to the stimulatory G-protein Gs leading to an increase in cAMP concentration in cells and a subsequent increase in the activation of PKA and PLC [10,11,12,13,14]. Among A2 subtypes, A2AR is related to ADO-induced anti-inflammatory and immune-suppressive functions in the TME. A2AR is highly expressed on diverse immune cells, such as lymphocytes, macrophages, neutrophils, dendritic cells, and natural killer (NK) cells [15]. Activation of A2AR suppresses the proliferation and effector functions of activated T cells through cAMP-PKA signaling pathways [16,17,18]. However, few reports have elucidated the role of A2AR in cancer cells and have shown conflicting results; stimulation of A2AR increased BC cell proliferation [19] but induced melanoma cell death via ERK signaling [20] and promoted colon cancer cell death via the caspase-9 and caspase-3 pathways [21]. However, the role of A2BR has been extensively studied in ADO-mediated tumor cell migration, invasion, and metastasis [22,23].

The radioresistance of BC cells remains a fundamental barrier to the maximum efficacy of radiotherapy. Therefore, in our previous study, we established radiotherapy-resistant (RT-R) BC cells and found that RT-R-TNBC exhibited more radio- and chemoresistance than non-TNBC cells; the cells also exhibited more aggressiveness in terms of tumor growth, invasion, and metastasis by regulating epithelial–mesenchymal transition (EMT)-related molecule expression and increasing the cancer stem cell population [24]. Interestingly, we found that P2Y_2_R expression was slightly decreased in RT-R-TNBC cells, while the expression of A2AR, A2BR, and CD73 was increased in RT-R-TNBC cells. Moreover, extracellular ATP and ADO levels were increased in RT-R-TNBC cells. Furthermore, our clinical study using BC patient specimens showed that the expression of A2AR, but not A2BR, was significantly increased in the tumor tissues, especially TNBC tissues, compared to the normal epithelial tissues of BC patients. Therefore, in this study, we investigated the role of ADO, its specific receptor A2AR, and the signaling pathway of A2AR in the progression of TNBC and RT-R-TNBC.

## 2. Materials and Methods

### 2.1. Case Selection

Specimens from 180 BC patients who underwent surgery with wide excision or mastectomy between January 2010 and December 2012 at Gyeongsang National University Hospital (Jinju, Korea) were selected. Each sample was fixed with formalin, embedded in paraffin, and stained with hematoxylin. The sections were prepared on glass slides and assessed by two pathologists. Data from electronic medical records, including sex, age, menstrual status, tumor size, lymph node status, distant metastasis, and tumor stage, were reviewed (Table 1). Cancer stages were determined according to the eighth edition of the American Joint Committee on Cancer (AJCC) staging system. Histological type and grade were determined according to the fifth edition of the World Health Organization (WHO) classification.

### 2.2. Tissue Microarray and Immunohistochemistry (IHC)

The prominent intratumoral regions from 180 BC patient specimens were chosen, and two 2-mm tissue cores were obtained from each paraffin block and transferred to recipient tissue microarray (TMA) blocks. Immunohistochemical staining was performed on the TMA blocks using anti-A2AR (ab3461, 1:100, Abcam, Cambridge, UK), anti-A2BR (ab229671, 1:100, Abcam), anti-CD73 (ab175396, 1:100, Abcam), and anti-CD39 (ab223842, 1:1000, Abcam) primary antibodies.

### 2.3. Evaluation of IHC Data

A2AR and CD73 membrane expression was analyzed by evaluating the staining intensity (0, negative; 1^+^, weak; 2^+^, moderate; 3^+^, strong). A2BR and CD39 were determined to be positive when staining in the cell membranes was detected in at least 10% of the cells.

### 2.4. Reagents and Cell Lines

Recombinant human TNF-α protein (210-TA-020) was purchased from R&D Systems (Minneapolis, MN, USA). ADO (A4036) was purchased from Sigma-Aldrich (St. Louis, MO, USA). The human BC cell lines MDA-MB-231, MCF-7, and T47D were obtained from the Korea Cell Line Bank (Seoul, Korea), and the human umbilical EC line EA.hy926 and mouse BC cell line 4T1 were obtained from the American Type Tissue Culture Collection (ATCC, Manassas, VA, USA). BC cells and ECs were grown in RPMI-1640 and DMEM, respectively, both of which were supplemented with 10% FBS (GenDEPOT, Katy, TX, USA) and 1% penicillin and streptomycin (HyClone; GE Healthcare Life Sciences, Logan, UT, USA).

### 2.5. Establishment of RT-R-BC Cells

RT-R-BC cells (RT-R-MDA-MB-231, RT-R-MCF-7, RT-R-T47D, and RT-R-4T1) were generated as described previously [24]. Briefly, BC cells (MDA-MB-231, MCF-7, T47D, and 4T1) were irradiated with fractionated X-ray irradiation (2 Gy) 25 times until a total of 50 Gy was reached. After cells were irradiated, they were grown up to ~90% confluence after changing the medium to a fresh one. Then, they were subcultured in new flasks. The cells were irradiated again when the confluence was ~70%. The RT-R-BC cells were used through 5 passages.

### 2.6. Total RNA Extraction and Reverse Transcription Polymerase Chain Reaction (RT-PCR)

Total RNA was extracted using TRIzol reagent (15596018, Thermo Fisher Scientific, Rockford, IL, USA). All primers were obtained from Bioneer, and RT-PCR was performed using TOPscript One-step RT-PCR Drymix (RT421, Enzynomics, Daejeon, Korea) according to the manufacturer’s instructions. The primer sets were as follows: hA1R, forward 5′-TCCCTCTCCGGTACAAGATG-3′ and reverse 5′-GCTGCTTGCGGATTAGGTAG-3; hA2AR, forward 5′-AGCTGAAGCAGATGGAGAGC-3′ and reverse 5′-AGGGATTCACAACCGAATTG-3; hA2BR, forward 5′-CAGCGGGAGATCCATGCAG-3′ and reverse 5′-CGGTTCCGGTAAGCATAGACAAT-3; hA3R, forward 5′-TACCCACGCCTCCATCATGT-3′ and reverse 5′-GGGGTCAATCCCACCAGGA-3; hCD39, forward 5′-CTGATTCCTGGGAGCACAT-3′ and reverse 5′-GACATAGGTGGAGTGGGAGAG-3; hCD73, forward 5′-GCCTGGGAGCTTACGATTTTG-3′ and reverse 5′-TAGTGCCCTGGTACTGGTCG-3; hGAPDH, forward 5′-TCAACAGCGACACCCACTCC-3′ and reverse 5′-TGAGGTCCACCACCCTGTTG-3. Thirty cycles of amplification were performed under the following conditions: melting at 95 °C for 30 s, annealing 57.5 or 60 °C for 30 s, and extension at 72 °C for 1 min.

### 2.7. Extracellular ADO and ATP Measurements

The extracellular level of ADO was measured with the Human ADO ELISA kit (MBS2605344, MyBioSource, San Diego, CA, USA) according to the manufacturer’s protocol. The extracellular level of ATP was measured as described previously [4] using the ENLITEN ATP Assay System kit (FF2000, Promega, Madison, WI, USA).

### 2.8. Cell Proliferation Assay

Cell proliferation was analyzed using the Cell Counting Kit-8 (CCK-8) assay. The cells were seeded in 96-well plates and treated with the indicated reagents for 24 h. Then, 10 µL/well of CCK-8 reagent (Dongin Biotech, Seoul, Korea) was added to the cells and incubated for 30 min, and the optical density (OD) of each well was measured at 450 nm using a microplate reader.

### 2.9. Gene Silencing by siRNA Transfection

The cells were transfected with 100 nM negative control siRNA (control siRNA, SN-1003, Bioneer, Daejeon, Korea) or A2AR siRNA (135-1, Bioneer) in serum-free medium using Lipofectamine 3000 (L300015, Thermo Fisher Scientific) for 4 h, and then the medium was replaced with fresh complete medium. The cells were starved with serum-free medium for 16 h and then treated with the indicated reagents. Gene silencing efficiency was determined by RT-PCR.

### 2.10. Colony Formation Assay

The cells (1 × 10^3^ cells) were seeded in 6-well plates and treated with the indicated reagents for 24 h. The culture medium was replaced with fresh complete medium every 2–3 days. After 1–2 weeks, the colonies were fixed in methanol for 10 min at room temperature and then stained with 0.1% Giemsa staining solution (32884, Sigma-Aldrich) diluted with distilled water, and the number of colonies was counted.

### 2.11. Matrigel Invasion Assay

The invasion assay was performed as described previously [4]. In brief, after harvesting the cells treated with the indicated reagents, 2 × 10^5^ cells were seeded in EC-Matrigel-coated upper chambers, and 500 μL of RPMI-1640 medium was added to the lower chambers. After 20 h, the noninvading cells remaining in the upper chamber were removed by scrubbing. The cells that invaded the insert membrane were stained with 4′,6-diamidine-2′-phenylindole dihydrochloride (DAPI, D8417, Sigma-Aldrich) and counted under a fluorescence microscope.

### 2.12. Protein Extraction and Western Blot Analysis

The cells were lysed in RIPA buffer as described previously [4]. Approximately 20–80 μg aliquots of protein were subjected to SDS–polyacrylamide gel electrophoresis and transferred onto PVDF membranes. The membranes were incubated with anti-p-AKT (9271, 1:1000, Cell Signaling, Danvers, MA, USA), anti-AKT (sc-8312, 1:1000, Santa Cruz Biotechnology, Dallas, TX, USA), anti-β-catenin (sc-7199, 1:2000, Santa Cruz Biotechnology), anti-Snail (3895, 1:1000, Cell Signaling), anti-vimentin (sc-6260, 1:1000, Santa Cruz), and anti-β-actin (A2066, 1:2000, Sigma-Aldrich) antibodies, and the bound antibodies were detected with horseradish peroxidase (HRP)-conjugated secondary antibodies and an enhanced chemiluminescence (ECL) Western blotting detection reagent (170-5061, Bio-Rad Laboratories, Hercules, CA, USA).

### 2.13. Animal Experiments

4T1 and RT-R-4T1 mouse BC cells were stably transfected with expression vectors encoding shRNAs targeting A2AR (sc-39751-SH, Santa Cruz Biotechnology) or empty vectors (EV, sc-108060, Santa Cruz Biotechnology). The subclones were designated 4T1-EV, 4T1-A2AR-shRNA, RT-R-4T1-EV, and RT-R-4T1-A2AR-shRNA. Six-week-old female BALB/c nude mice were injected subcutaneously with 1 × 10^5^ cells/100 μL serum-free medium from each of the four subclones. Body weight and tumor volume were measured twice a week, starting 7 days after injection. On the 24th day, the mice were sacrificed, and the tumor and lung tissues were extracted. The tumor tissues were fixed in 10% formalin at room temperature and then subjected to paraffin infiltration and embedding. Sections of 5 μm were mounted onto glass slides, and immunohistochemical analysis was performed using the following primary antibodies: anti-A2AR (ab3461, 1:100, Abcam), anti-p-AKT (ab81283, 1:100, Abcam), anti-β-catenin (ab16051, 1:100, Abcam), anti-Snail (ab180714, 1:100, Abcam), and anti-vimentin (sc-6260, 1:50, Santa Cruz). Then, the sections were incubated with HRP-conjugated secondary antibodies, and IHC staining was performed using ABC solution (Vector Labs, Burlingame, CA, USA) and diaminobenzidine (DAB) according to the manufacturers’ instructions. After color development, counterstaining was performed with Mayer’s hematoxylin, and the cells were observed under a light microscope (CKX41, Olympus, Tokyo, Japan). The animal experiment protocol was approved by the Institutional Animal Care and Use Committee at Gyeongsang National University (approval number: GNU-200603-M0030), and all experiments were conducted in compliance with the institutional guidelines.

### 2.14. Statistical Analysis

All data were statistically analyzed by using GraphPad Prism 7 software. One-way ANOVA followed by Tukey’s post hoc test was carried out to compare different groups. The data are presented as the mean ± SD. Correlation analyses were performed using the chi-square test and Fisher’s exact test.

## 3. Results

### 3.1. The Expression of A2AR, A2BR, and CD73 Is Increased in TNBC and Further Increased in RT-R-TNBC Cells

First, to clarify the role of purinergic receptors in RT-R-BC cells, we observed the mRNA expression levels of purinergic receptors, including P2Y_2_R and ADO receptors, in various BC cells (MDA-MB-231, MCF-7, T47D) and RT-R-BC cells (RT-R-MDA-MB-231, RT-R-MCF-7, RT-R-T47D), with the normal epithelial cell line MCF-10A as the control. Unexpectedly, P2Y_2_R expression was slightly decreased in RT-R-MDA-MB-231 cells compared to MDA-MB-231 cells, while A2AR, A2BR, and CD73 expression levels were higher in TNBC cell lines than in non-TNBC cell lines. A2AR mRNA expression was induced in RT-R-BC cells in response to TNF-α treatment (Figure 1).

### 3.2. Control- and TNF-α-Treated MDA-MB-231 Cells Show Higher Extracellular ATP and ADO Levels Than Non-TNBC Cells, and This Effect Is Further Enhanced in RT-R-MDA-MB-231 Cells

Then, we examined the production of ATP and ADO in BC cells and RT-R-BC cells with or without TNF-α stimulation. Similar to a previous study [25], our study showed that RT-R-MDA-MB-231 cells displayed the highest basal ATP level among all RT-R-BC cells and exhibited significantly increased ATP levels in response to TNF-α stimulation, with a peak at 5 min after TNF-α treatment (Figure 2A). RT-R-MDA-MB-231 cells also showed the highest ADO level among BC cells and RT-R-BC cells. Interestingly, the ADO levels peaked at 10 min after TNF-α treatment in RT-R-MDA-MB-231 and MDA-MB-231 cells, suggesting the possibility of the conversion of ATP to ADO (Figure 2B).

### 3.3. The Expression of A2AR, but Not A2BR, CD39, and CD73, Is Significantly Increased in BC Patient Tissues, Especially TNBC Patient Tissues, Compared to Normal Epithelial Tissues

To determine the clinical importance of A2AR, A2BR, and CD73 in breast tumor patients, we examined the expression of these molecules in tumor tissues (*n* = 180) and normal epithelial tissues (*n* = 20) obtained from BC patients. Interestingly, the expression of A2AR, but not A2BR, CD39, and CD73, was significantly increased in the tumor tissues compared to the normal epithelial tissues of BC patients (Figure 3A). Moreover, when we compared the expression levels of these proteins between TNBC tumor tissues (*n* = 20) and non-TNBC tumor tissues (*n* = 160), the A2AR expression level was significantly higher in TNBC tumor tissues (Figure 3B). Figure 3C shows a representative image of the immunohistochemical staining of A2AR between tumor tissues and normal tissues (Figure 3C). As shown in Table 1, A2AR, but not A2BR, was significantly related to the TNBC subtype. These results suggest that extracellular ADO can be related to the clinicopathological characteristics of BC patients through A2AR. Thus, we focused on the role of A2AR in the tumor progression of BC cells, especially in RT-R-TNBC, and assessed the possible mechanisms.

### 3.4. Extracellular ADO Enhances the Proliferation, Colony Formation, and Invasion of MDA-MB-231 and RT-R-MDA-MB-231 Cells through A2AR Activation

We examined the effect of ADO on the proliferation of both MDA-MB-231 and RT-R-MDA-MB-231 cells. As shown in Figure 4A, RT-R-MDA-MB-231 cells showed higher proliferation levels than MDA-MB-231 cells, and ADO treatment significantly increased the proliferation of RT-R-MDA-MB-231 cells in a dose-dependent manner (1–200 μM). MDA-MB-231 cells also exhibited increased proliferation after ADO treatment in the dose range of 1–100 μM, with a slight decrease at 200 μM. The increase in cell proliferation in both BC cell lines was notably suppressed by A2AR siRNA transfection (Figure 4B). Moreover, knockdown of A2AR dramatically attenuated the ADO-treatment-induced increases in the colony-forming abilities of MDA-MB-231 and RT-R-MDA-MB-231 cells (Figure 4C,D). Furthermore, RT-R-MDA-MB-231 cells displayed a higher invasive ability than MDA-MB-231 cells, and ADO treatment markedly enhanced the invasion of MDA-MB-231 and RT-R-MDA-MB-231 cells; however, this effect was markedly attenuated by A2AR siRNA transfection (Figure 4E,F).

### 3.5. Extracellular ADO-Mediated Activation of A2AR Stimulates AKT Activation and Induces β-Catenin, Snail, and Vimentin Expression, Which Are Involved in Tumor Invasion and Metastasis

Next, we wondered what signaling molecules are linked to activated A2AR signaling. EMT is the process by which epithelial cells change their morphology to a mesenchymal shape and finally acquire invasive and metastatic capacity [26,27]. In addition, Shi et al. recently reported that A2AR activation promotes gastric cancer metastasis by inducing PI3K-AKT-mTOR signaling [28]. Therefore, we investigated whether A2AR activation is related to the PI3K-AKT pathway and/or EMT-related signaling. RT-R-MDA-MB-231 cells showed higher expression levels of p-AKT, β-catenin, and Snail than MDA-MB-231 cells, whereas their vimentin expression level was similar to that of MDA-MB-231 cells and very weak compared to other EMT proteins such as β-catenin and Snail. Interestingly, ADO treatment induced AKT phosphorylation and β-catenin, Snail, and vimentin expression in both BC cell lines; RT-R-MDA-MB-231 cells showed an earlier induction and longer maintenance of the expression of β-catenin and Snail and an induction of vimentin at a later time (at 24 h) than MDA-MB-231 cells (Figure 5A,B; Appendix A). The ADO-induced increases in p-AKT and EMT-related molecules (β-catenin, Snail, and vimentin) levels were attenuated by A2AR siRNA transfection (Figure 5C–F, Appendix A). These results suggest that extracellular ADO induces AKT activation and consequent β-catenin, Snail, and vimentin expression in MDA-MB-231 and RT-R-MDA-MB-231 cells in an A2AR-dependent manner, resulting in the promotion of the EMT process and tumor invasion.

### 3.6. A2AR Is Involved in the Progression and Metastasis of BC In Vivo

Finally, we confirmed the role of A2AR in breast tumor growth and metastasis using an in vivo allograft mouse model. For in vivo animal experiments, we constructed stable A2AR knockdown cell lines using A2AR shRNA or empty vector (EV) (Figure 6A). Next, we divided mice into four groups (4T1-EV, 4T1-A2AR-shRNA, RT-R-4T1-EV, RT-R-4T1-A2AR-shRNA) and injected them with these BC cells. When we measured tumor volume and body weight as described in Section 2, the tumor volumes were found to be significantly increased in the RT-R-4T1-EV group compared to the 4T1-EV group and notably reduced in the 4T1-A2AR-shRNA and RT-R-4T1-A2AR-shRNA groups compared to the EV group (Figure 6B–D). The body weights during the experimental period were similar among the four groups (Figure 6E); however, metastasis to the lung was remarkably increased in the RT-R-4T1-EV group compared to the 4T1-EV group and was decreased in the RT-R-4T1-A2AR-shRNA group (Figure 6F). Finally, immunohistochemical analysis showed that the β-catenin and Snail expression levels were higher in the RT-R-4T1-EV group than in the 4T1-EV group and were reduced in the 4T1-A2AR-shRNA and RT-R-4T1-A2AR-shRNA groups (Figure 6G). Interestingly, unlike what was observed for human BC cell lines MDA-MB-231 and RT-R-MDA-MB-231, in vivo data showed higher expression levels of vimentin in RT-R-4T1-VE than 4T1-EV group, which were reduced in the 4T1-A2AR-shRNA and RT-R-4T1-A2AR-shRNA groups. The expression of p-AKT was hardly detected in the tumor tissues extracted from the mice of each group but was relatively high in the RT-R-4T1-EV group compared to the other groups. These results suggest that A2AR is related to tumor growth and metastasis in BC, especially in RT-R-TNBC.

## 4. Discussion

In the TME, diverse cytokines and chemokines accumulate and stimulate signal transduction in adjacent cells, including cancer cells. Among these stimuli, TNF-α is abundant and plays a key role in angiogenesis, apoptosis, inflammation, and immunity in the TME [29]. Accumulating evidence shows that TNF-α contributes to tumor initiation and progression [30,31,32,33]. Increased TNF-α blood levels are observed in cancer patients and are correlated with an increased risk of metastatic diseases [34]. Moreover, under these conditions, ATP is secreted from highly metastatic TNBC cells, accumulates at a high concentration, and plays an important role in tumor progression and metastasis by regulating crosstalk between other cells through P2Y_2_R [4,35]. To further clarify the role of ATP and P2Y_2_R in RT-R-TNBC cells, we assessed P2Y_2_R expression and interestingly found that P2Y_2_R expression was slightly decreased in RT-R-MDA-MB-231 cells compared to MDA-MB-231 cells, while the expression of A2AR, A2BR, and CD73 was increased in TNBC and was further increased in RT-R-TNBC cells (Figure 1). Moreover, extracellular ADO and ATP levels were increased in RT-R-MDA-MB-231 cells (Figure 2). According to a recent study, abundant extracellular ADO also regulates the properties and functions of various cells, including tumor cells and immune cells in the TME, through P1 receptors [3]. Therefore, we hypothesized that extracellular ADO may play a more important role in tumor progression in RT-R-TNBC cells through P1 receptors.

To date, the role of A2AR has been studied mostly in immune cells such as lymphocytes, macrophages, neutrophils, dendritic cells, and NK cells; A2AR is reported to suppress immune activities when it is activated by ADO [36]. However, accumulating evidence indicates that A2BR is expressed in various cancers and is involved in tumor progression [37]. According to Fernandez-Gallardo et al. [38], A2BR is the most highly expressed receptor, whereas A2AR is hardly detected in MDA-MB-231 cells. However, in our study, as shown in Figure 1, the A2AR expression level was high and was further increased in response to TNF-α in MDA-MB-231 and RT-R-MDA-MB-231 cells. The expression level of A2BR was the highest in MDA-MB-231 cells but did not change in response to TNF-α, even in RT-R-MDA-MB-231 cells. A1R and A3R showed low expression levels in all BC cells, including TNBC, non-TNBC, and RT-R-BC cells, and in normal epithelial cells. Emerging studies have indicated that CD73 is a key regulator of cancer cell proliferation, migration, invasion, angiogenesis, and tumor immune escape [39]. Moreover, the expression of CD73 is associated with a poor prognosis in diverse cancers, including brain cancer [40], prostate cancer [41,42], ovarian cancer [43,44], BC [45], and leukemia [46,47]. Moreover, chemotherapy resistance has been reported in CD73-expressing cancers [40,45,47]. However, the tumor progression-related functions of CD73 are not self-induced but are ultimately exerted through P1 receptors, especially A2AR and A2BR, which are activated by extracellular ADO, which is produced by CD73 [3]. Therefore, we postulated that highly expressed CD73 is involved in the increase in extracellular ADO from ATP, which is released from TNBC cells in large amounts, and we focused on the role of the ADO receptor on RT-R-BC cells. When we analyzed the expression of A2AR and A2BR in the tumor tissue and normal tissue of BC patients, A2AR expression, but not A2BR expression, was increased in tumor tissues, especially TNBC tissues, compared to normal epithelial tissues (Figure 3A,B; Table 1). Based on these results, we suggest that the role of A2AR might be more important and relevant to clinical cancer progression than that of A2BR.

As mentioned before, EMT is the fundamental biological process by which tumor cells acquire invasive phenotypes through the loss of epithelial characteristics and the acquisition of mesenchymal characteristics, and this process plays an important role in the metastasis of many cancers, including BC [26]. Snail, Slug, Twist, and ZEB, which are well-known key transcription factors for EMT, are activated by Wnt/β-catenin signaling, EGF/FGF–RTK signaling, Notch signaling, MAPK signaling and TGFβ–SMAD signaling. After activation of these signaling pathways, alterations in EMT-related gene expression are initiated; the expression levels of E-cadherin and ZO1 are decreased, whereas those of N-cadherin, integrins, fibronectin, vimentin, and MMPs are increased [27]. Moreover, these transcription factors are overexpressed in primary invasive BC and related to a poor prognosis [48,49,50]. Snail is known as a direct target of Wnt/β-catenin signaling in BC [51,52], and AKT is a core signaling pathway with over a hundred downstream target substrates that regulate growth, survival, proliferation, and cell metabolism [53,54]. Furthermore, the overexpression of p-AKT is a general feature in early and advanced tumors [54]. Increased p-AKT is observed in various cancers, including BC [55,56,57], and during melanoma formation and progression [58]. The roles of EMT in cell invasion and migration, tumor recurrence, and chemotherapy resistance have been investigated intensively [59,60,61,62,63], and recent emerging evidence suggests that EMT also plays a determinant role in the development of radioresistance in cancer cells [64,65,66,67]. However, the role of EMT in the development of BC radioresistance and the involvement of ADO-A2AR in the EMT and metastasis of radioresistant BC cells remain unclear. In our previous study, we determined that the expression levels of EMT-related molecules, Snail, β-catenin, and N-cadherin, are increased in RT-R-MDA-MB-231 cells compared with MDA-MB-231 cells, indicating that the RT-R cells have more invasive features [24]. Therefore, in this study, we investigated the role of A2AR in the regulation of EMT-related molecules and its relationship with AKT signaling. The results showed that RT-R-MDA-MB-231 cells, which produced more ADO than MDA-MB-231 cells, exhibited increased AKT phosphorylation and β-catenin and Snail expression (Figure 5). Moreover, extracellular ADO induced AKT phosphorylation and consequent β-catenin and Snail expression in an A2AR-dependent manner in RT-R-MDA-MB-231 and MDA-MB-231 cells. In addition, vimentin, which showed similar expression levels between MDA-MB-231 and RT-R-MDA-MB-231 cells also induced in response to extracellular ADO, especially exhibited highly augmented expression in RT-R-MDA-MB-231 cells in an A2AR-dependent manner (Figure 5). Although we examined the expression of other EMT-related molecules, such as E-cadherin, N-cadherin, and fibronectin, in MDA-MB-231 and RT-R-MDA-MB-231 cells, we could not detect alterations in response to ADO.

## 5. Conclusions

Taken together, our present results reveal for the first time the role of A2AR in the progression and metastasis of RT-R-BC cells as well as BC cells. We are the first to suggest that the ADO-activated intracellular A2AR signaling pathway is linked to the AKT-β-catenin pathway to regulate BC cell invasiveness and metastasis to the lung. Because TNBC is a difficult cancer type to treat and RT-R is a major obstacle to complete cancer therapy, it might be helpful to investigate the role of A2AR in TNBC and RT-R-TNBC.

## Figures and Tables

**Figure 1 cancers-13-02105-f001:**
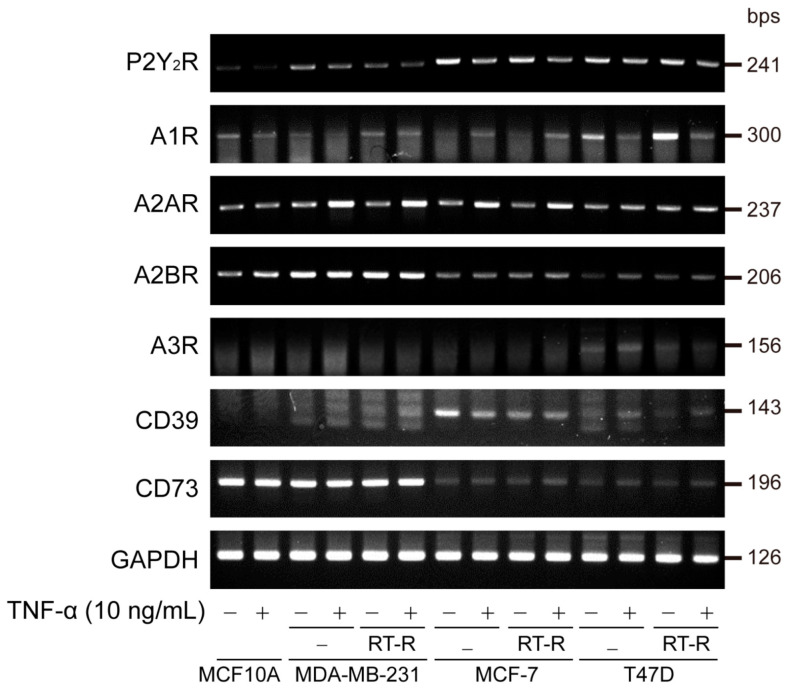
Expression levels of purinergic P2Y_2_ receptor (P2Y_2_R), adenosine receptors (A1R, A2AR, A2BR, A3R), and ectonucleotidases (CD39, CD73) in various breast cancer (BC) cells and their radiotherapy-resistant (RT-R)-BC cells. Normal epithelial cell MCF-10A, triple-negative breast cancer cell (TNBC) MDA-MB231, non-TNBCs MCF-7 and T47D, and their RT-R-BC cells were treated with or without TNF-α for 24 h. Total RNA was collected from the cells, and P2Y_2_R, adenosine receptor (A1R, A2AR, A2BR, A3R), ectonucleotidases (CD39, CD73), and GAPDH mRNA expression levels were analyzed by RT-PCR as described in Section 2. Results were confirmed by repeated experiments.

**Figure 2 cancers-13-02105-f002:**
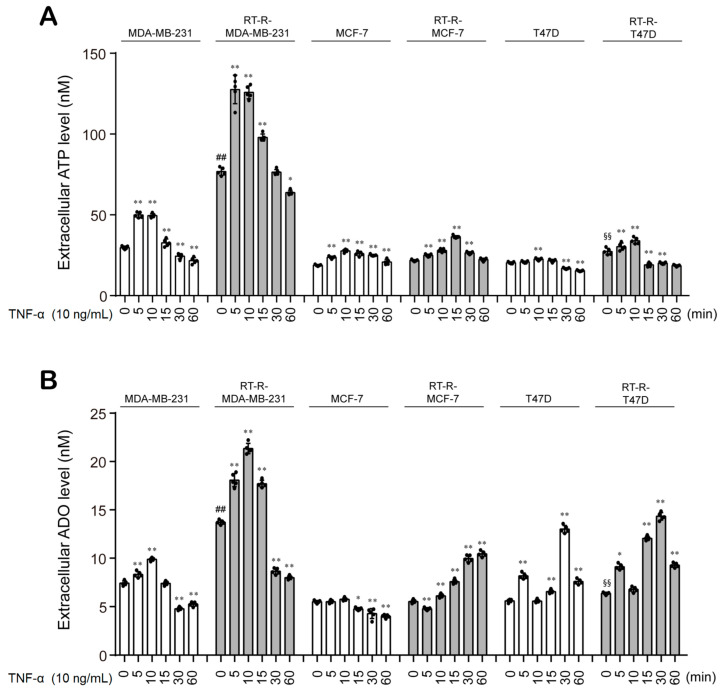
The extracellular levels of ATP and ADO are increased in the MDA-MB-231, a TNBC cell line, when compared to any other non-TNBC cells, and are more enhanced in RT-R-MDA-MB-231 cells. (**A**,**B**) BC cells (MDA-MB-231, MCF7, T47D) and their RT-R-BC cells (RT-R-MDA-MB-231, RT-R-MCF7, RT-R-T47D) were treated with TNF-α for the indicated time, and the extracellular levels of ATP (**A**) and ADO (**B**) were measured using the ATP assay system kit and ADO ELISA Kit, respectively, as described in Section 2. The values represent the means ± SD of 5 independent experiments. * *p* < 0.05, ** *p* < 0.01 compared to the control of each cell line; ^##^
*p* < 0.01 compared to the control of MDA-MB-231; ^§§^
*p* < 0.01 compared to the control of T47D.

**Figure 3 cancers-13-02105-f003:**
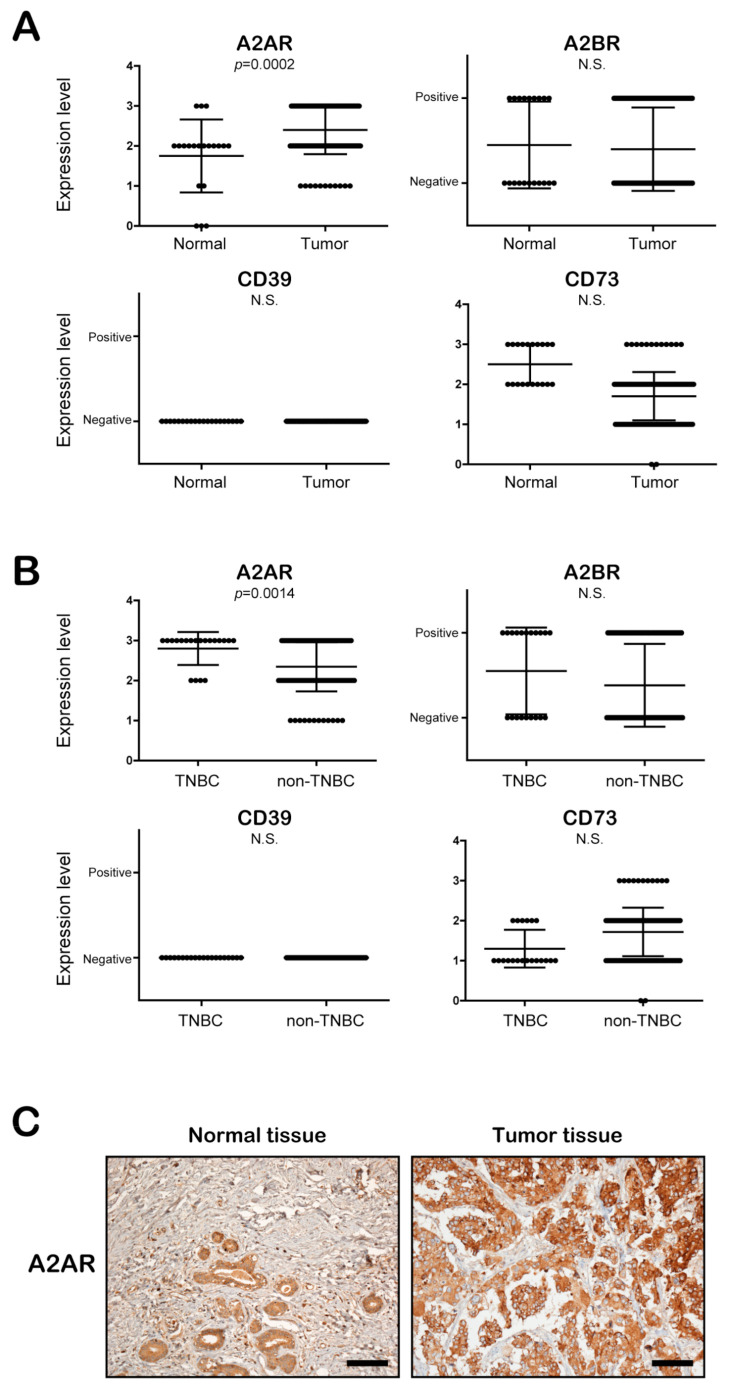
A2AR expression is significantly increased in the tumor tissue, especially in TNBC, compared to normal epithelial tissues of breast cancer patients. (**A**,**B**) The expression levels of A2AR, A2BR, CD39, and CD73 were evaluated in normal epithelial tissues (*n* = 20) and tumor tissues (*n* = 180) (**A**) and in TNBC tissues (*n* = 20) and non-TNBC tissues (*n* = 160) (**B**) of patients with breast cancer. (**C**) Immunohistochemical staining of A2A2R in tumor tissues and normal epithelial tissues of breast cancer patients (scale bar, 100 μm).

**Figure 4 cancers-13-02105-f004:**
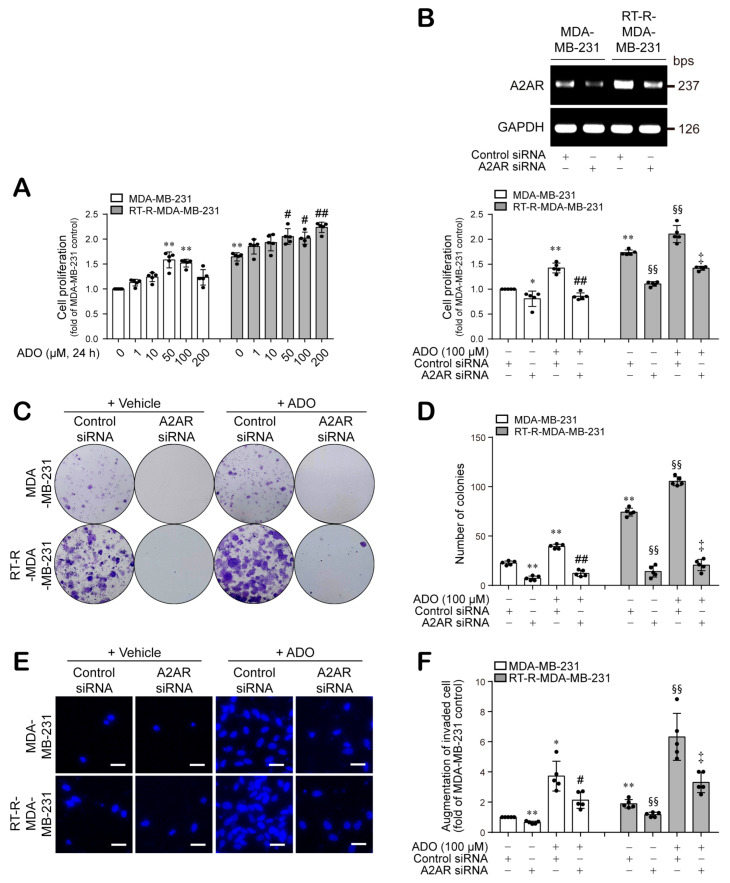
Extracellular ADO increased cell proliferation and colony-forming ability of MDA-MB-231 and RT-R-MDA-MB-231 cells in an A2AR-dependent manner. (**A**) Cells were treated with ADO at indicated doses for 24 h, and the cell proliferation was measured by CCK-8 assay as described in Section 2. (**B**) Cells were transfected with negative control (CTRL) siRNA (100 nM) or A2AR siRNA (100 nM) for 4 h, and then the medium was replaced with fresh complete medium. After 24 h, total RNA was collected, and the efficiency of A2AR siRNA on A2AR expression was determined by RT-PCR. Then, CTRL siRNA- or A2AR siRNA-transfected (100 nM) cells were treated with ADO (100 μM) for 24 h, and then cell proliferation was measured as above. (**C,D**) CTRL siRNA- or A2AR siRNA-transfected cells were treated with ADO (100 μM) for 24 h, and then culture medium was discarded and replaced with fresh complete medium every 2–3 days. After 1–2 weeks, colony-forming ability was determined as described in Section 2. (**E,F**) Cells transfected with indicated siRNAs were treated with ADO (100 μM) for 24 h, and the invasion assay was performed and quantified as described in Section 2. The values represent the means ± SD of 5 independent experiments (scale bar, 100 μm). * *p* < 0.05, ** *p* < 0.01 compared to the control of MDA-MB-231; ^#^
*p* < 0.05, ^##^
*p* < 0.01 compared to the ADO treatment of MDA-MB-231; ^§§^
*p* < 0.01 compared to the control of RT-R-MDA-MB-231; ^‡^
*p* < 0.01 compared to the ADO treatment of RT-R-MDA-MB-231.

**Figure 5 cancers-13-02105-f005:**
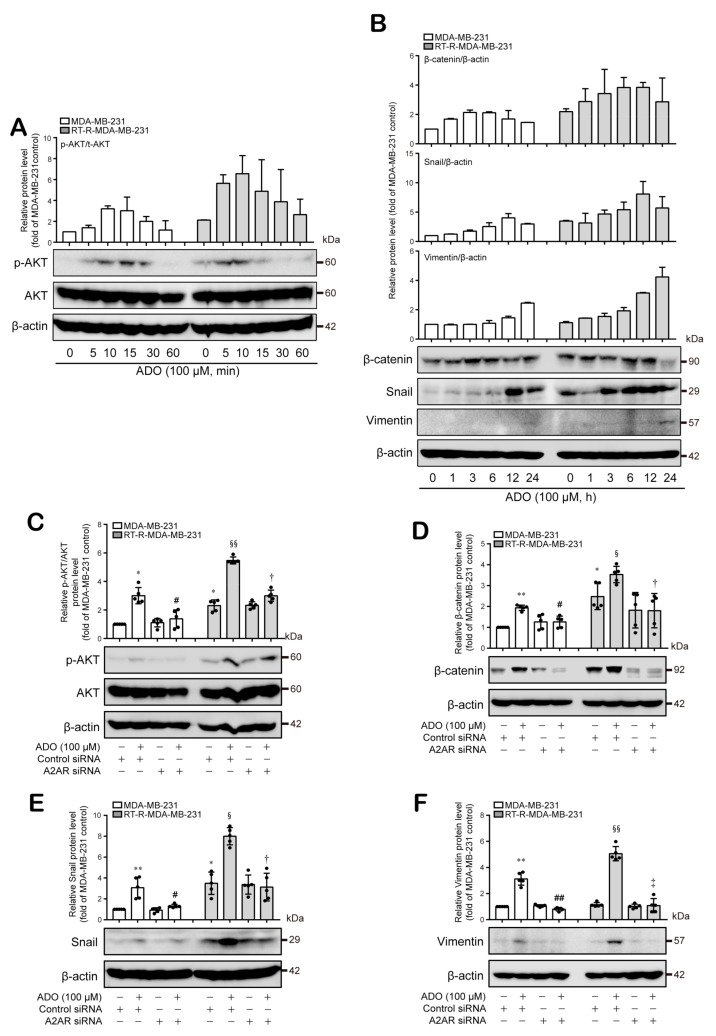
Extracellular ADO activated AKT pathway and mediated β-catenin, Snail, and vimentin expression in MDA-MB-231 and RT-R-MDA-MB-231 cells. (**A,B**) Cells were time-dependently treated with ADO (100 μM), and the p-AKT, AKT (**A**), β-catenin, Snail, vimentin, and β-actin protein levels (**B**) were analyzed by Western blotting. The results were confirmed by repeated experiments. (**C**–**F**) Cells transfected with CTRL siRNA or A2AR siRNA were treated with ADO for 10 min (AKT), 6 h (β-catenin), 12 h (Snail), and 24 h (vimentin) for the detection of each molecule. Then, cells were harvested, and the p-AKT, AKT, β-catenin, Snail, vimentin, and β-actin protein levels in the lysates were analyzed by Western blotting, and the band intensities were assessed by scanning densitometry. The values were represented as the means ± SD of 5 independent experiments. * *p* < 0.05, ** *p* < 0.01 compared to the control of MDA-MB-231; ^#^
*p* < 0.05, ^##^
*p* < 0.01 compared to the ADO treatment of MDA-MB-231; ^§^
*p* < 0.05, ^§§^
*p* < 0.01 compared to the control of RT-R-MDA-MB-231; ^†^
*p* < 0.05, ^‡^
*p* < 0.01 compared to the ADO treatment of RT-R-MDA-MB-231.

**Figure 6 cancers-13-02105-f006:**
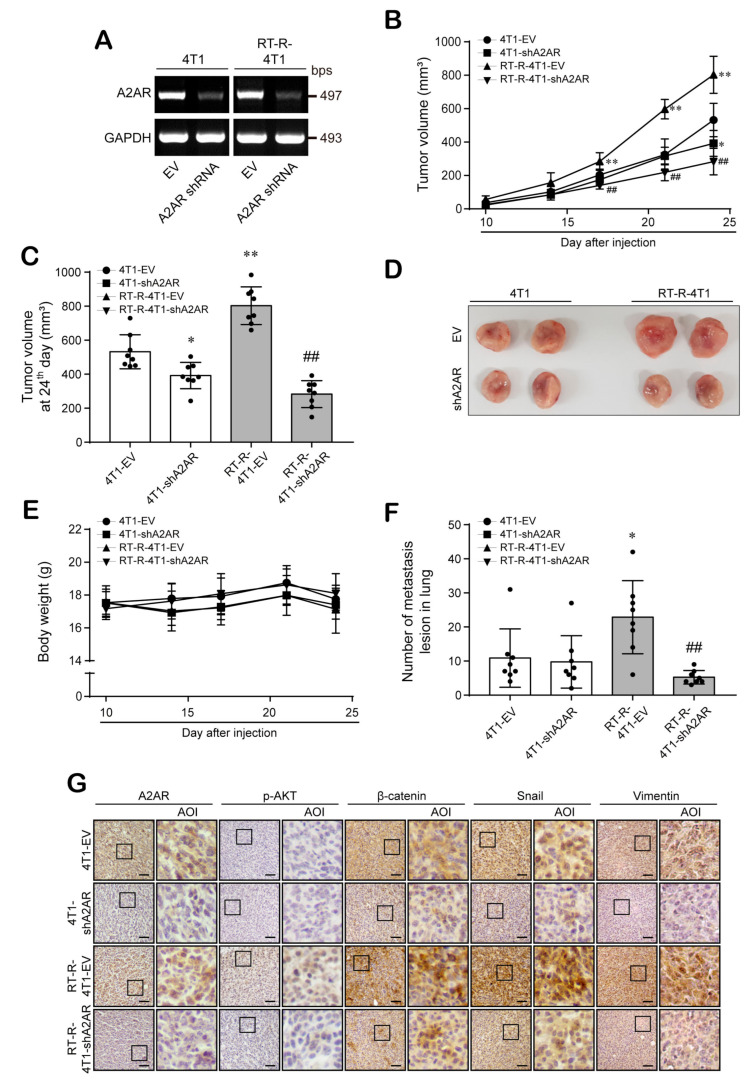
A2AR plays an important role in tumor growth and metastasis in in vivo mouse models. (**A**) Mouse breast cancer cells 4T1 and its RT-R-cells (RT-R-4T1) were stably transfected with empty vector (100 nM; 4T1-EV and RT-R-4T1-EV) or A2AR shRNA (100 nM; 4T1-shA2AR and RT-R-4T1-shA2AR) as described in Section 2. (**B**–**F**) Balb/c nude mice were divided into 4 groups (*n* = 8/each group) and injected subcutaneously (1 × 10^5^ cells/100 µL of serum-free medium). The animals were sacrificed on the 24th day, and the tumors and lung tissues were extracted. Tumor volumes (**B**–**D**) and body weights (**E**) were measured twice a week during tumor development. The incidence of lung metastasis (**F**) was investigated after sacrifice. * *p* < 0.05, ** *p* < 0.01 compared to 4T1-EV; ^##^
*p* < 0.01 compared to RT-R-4T1-EV. (**G**) Tumor tissue sections were stained with anti-A2AR, anti-p-AKT, anti-β-catenin, anti-Snail, and anti-vimentin antibodies as described in Section 2 (scale bar, 50 µm), and the sections were counterstained with Mayer’s hematoxylin solution. EV, empty vector; AOI, area of interest.

**Table 1 cancers-13-02105-t001:** Association between P1 receptors, A2AR and A2BR, and clinicopathological characteristics in breast cancer patients.

Characteristics	A2AR		*p*	A2BR		*p*
	Low (*n* = 97)	High (*n* = 83)		Negative (*n* = 108)	Positive (*n* = 72)	
Age (years)	51 (30–81)	51 (25–82)	0.637	51 (30–81)	51 (25–82)	0.699
Sex			0.461			0.400
Female	97 (100.0)	82 (99.8)		108 (100.0)	71 (98.6)	
Menopausal status			0.651			>0.990
Pre	44 (45.4)	34 (41.0)		47 (43.5)	31 (43.1)	
Post	53 (54.6)	48 (57.8)		61 (56.6)	40 (55.6)	
Histology			0.055			0.785
Ductal	93 (95.9)	73 (88.0)		99 (91.7)	67 (93.1)	
Others	4 (4.1)	10 (12.0)		9 (8.3)	5 (6.9)	
ER status			<0.001			0.093
Negative	14 (14.4)	36 (43.4)		25 (23.1)	25 (34.7)	
Positive	83 (85.6)	47 (56.6)		83 (76.9)	47 (65.3)	
PR status			0.133			0.234
Negative	22 (22.7)	28 (33.7)		34 (31.5)	16 (22.2)	
Positive	75(77.3)	55 (66.3)		74 (68.5)	56 (77.8)	
HER-2 status			0.685			0.543
Negative	80 (82.5)	71 (85.5)		89 (82.4)	62 (86.1)	
Positive	17 (17.5)	12 (14.5)		19 (17.6)	10 (13.9)	
TNBC			0.002			0.155
Yes	4 (4.1)	16 (19.3)		9 (8.3)	11 (15.3)	
No	93 (95.9)	67 (80.7)		99 (91.7)	61 (84.7)	
AJCC 8th stage			0.135			0.916
I	33 (34.0)	37 (44.6)		42 (38.9)	28 (38.9)	
II	47 (48.5)	28 (33.7)		46 (42.6)	29 (40.3)	
III	17 (17.5)	18 (21.7)		20 (18.5)	15 (20.8)	
Surgery			0.532			0.341
MRM	32 (33.0)	32 (38.6)		35 (32.4)	29 (40.3)	
BCS	65 (67.0)	51 (61.4)		73 (67.6)	43 (59.7)	
Adjuvant chemotherapy			0.806			0.801
Yes	88 (90.7)	74 (89.2)		98 (90.7)	64 (88.9)	
No	9 (9.3)	9 (10.8)		10 (9.3)	8 (11.1)	
Adjuvant radiotherapy			0.608			0.119
Yes	74 (76.3)	60 (72.3)		85 (78.7)	49 (68.1)	
No	23 (23.7)	23 (27.7)		23 (21.3)	23 (31.9)	
Adjuvant hormone therapy			0.018			0.521
Yes	89 (91.8)	65 (78.3)		94 (87.0)	60 (83.3)	
No	8 (8.2)	18 (21.7)		14 (13.0)	12 (16.7)	

## Data Availability

All data are available via the corresponding author.

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
