# Peer review of "Increased Extracellular Adenosine in Radiotherapy-Resistant Breast Cancer Cells Enhances Tumor Progression through A2AR-Akt-β-Catenin Signaling"

_cancers, 2021, doi:10.3390/cancers13092105_

Round 1

Reviewer 1 Report

The authors have satisfied all major concerns.  

I have a concern about the quality of the western blot data provided in figure 5.  Fig 5B; it seems like Vimentin didn't detect in untreated MDA-MA-231! Since authors performed 5 independent experiments. I would suggest quantifying the WB of replicates. 

Author Response

->Answer: Thank you for your time and comment. I fully understand your concern. As you see from Figure 5B, the level of Vimentin was very weak in both untreated MDA-MB-231 and RT-R-MDA-MB-231 cells compared to other EMT proteins such as β-catenin and Snail, and increased in response to ADO treatment at later time (at 24 h). As we described in the Figure 5 legends (lines 33, 339), we confirmed the time dependency (Figure 5A, B) by repeated experiments. Thereafter, in Figure 5C~F, we performed the 5 independent experiments and did statistical analysis (please see the quantifying graph for the WB in Figure 5C~F).

During revision, as you suggested, we included quantified graph in Figure 5A and B, even though it is not possible to do statistical analysis. In addition, we corrected some description for the Vimentin result (please Figure 5A and B; lines 324, 325, 328).

Thank you again for your comment.

This manuscript is a resubmission of an earlier submission. The following is a list of the peer review reports and author responses from that submission.

Round 1

Reviewer 1 Report

The current manuscript entitled, "Increased Extracellular Adenosine in Radiotherapy-Resistant Breast Cancer Cells Enhances Tumor Progression through A2AR-Akt-β-catenin Signaling" well-articulated the role of adenosine in cancer progression in radiation-resistant Breast cancer by activating its receptor A2AR and its downstream signaling molecules. Though, the data presented by the authors is quite interesting, which is nicely conducted and can be accepted for publication. I have following questions to ask;
Authors have generated radiation resistance BC cells in vitro, however, In the manuscript, authors haven't shown any data to demonstrate or compared the effects of combined treatment of irradiation alone and with ADO or THF-a on parental and/or radiation resistance cells? 
It is known that ionization induces double-strand DNA break. did the author observe any difference in DNA damage change in the presence and absence of ADO? 
In the current manuscript, the authors haven't presented data to support metastasis. what type of metastasis authors referred to in conclusion ?. 
Radiation resistant cells are known to acquire antioxidant potential thereby regulate mitochondria and ameliorates ATP production. Did the authors observe any intracellular redox change upon TNFa stimulation in RR-R-MDA-MB-231 cells? 

Reviewer 2 Report

The tumor microenvironment (TME) plays an essential role in tumor progression. It has been shown, that components of TME interacts with cancer cells which can result in metastasis, increasing invasion potential, and radio- or chemotherapy resistance. In this article authors analyze the role of adenosine (present in TME) and adenosine receptors present on breast cancer cells, as well as A2AR signaling pathway in the progression of triple negative breast cancer cells.

Although the role of adenosine (ADO) in cancer has been extensively studied, the aspect of  ADO impact on radiotherapy resistant breast cancer cells in new, interesting and worth studying.

The paper is well written and easy to read. Authors addressed the raised hypothesis, however the excessive conclusions has been drawn, as the experiments were conducted only on one, very aggressive and prone to metastasis TNBC cell line. As the TNBC is also highly heterogeneous (for example p[resenting epithelial or mesenchymal features), more diverse cell lines should be used to confirm obtained results.

Major concerns:

Most doubts in this publication are raised by only one TNBC cell line on which experiments were performed. Breast cancer is a very heterogeneous cancer, with different molecular characteristics, and even the TNBC shares more mesenchymal-like, basal-like or luminal morphology and this also should be taken into account. The most important conclusions should also by validated on another TNBC cell lines and primary cells isolated from patients.

Fig. 4 There are no experiment showing untreated cells. This should be added to see if siRNA affects the cells. For example if almost no colonies are present in control siRNA well (4C) it means that control siRNA significantly affected MDA-MB-231 proliferation. It is almost impossible that so few colonies were present in siRNA control treated cells. Moreover, images showing bigger area of the well should be added, as the colonies in 4C do not represent graphs on 4D. Moreover, information should be added on how many cells in colony were assumed to be a colony. Fig. 4E should be in higher resolution.

Fig. 5 Moreover studying the expression of only 2 genes (beta-catenin and Snail) is insufficient to define EMT process, tumor invasion and metastasis. Some additional markers, which are characteristic for EMT in breast cancer should be added (Vimentin, N-cadherin, alpha-SMA). Moreover, looking more closely on losing the epithelial characteristics should also be important. Moreover as mentioned above, studies only on one cell line with a mesenchymal characteristic is insufficient to draw conclusions about the impact of ADO-mediated activation of A2AR and AKT signaling pathway on TNBC tumor invasion and metastasis.

Minor concerns:

Authors have already published some papers concerning this topic, and this publication is a good continuation of previously published papers. However, as a result, authors too often refer to their previous publications, making it difficult to follow the mainstream of publications due to the need to seek information from their previous publications. The most important aspect, is to mention in introduction about obtaining RT-R-TNBC cells. With clarifying their characteristics, it would be easier to understand the manuscript.